# Efficacy of Cisplatin–CXCR4 Antagonist Combination Therapy in Oral Cancer

**DOI:** 10.3390/cancers16132326

**Published:** 2024-06-25

**Authors:** Saori Yoshida, Hotaka Kawai, Yamin Soe, Htoo Shwe Eain, Sho Sanou, Kiyofumi Takabatake, Yohei Takeshita, Miki Hisatomi, Hitoshi Nagatsuka, Junichi Asaumi, Yoshinobu Yanagi

**Affiliations:** 1Preliminary Examination Room, Okayama University Hospital, Okayama 700-8558, Japan; pbfo02xb@s.okayama-u.ac.jp (S.Y.); ya7@okayama-u.ac.jp (Y.Y.); 2Department of Oral Pathology and Medicine, Graduate School of Medicine, Dentistry and Pharmaceutical Sciences, Okayama University, Okayama 700-8525, Japan; pki31ld5@s.okayama-u.ac.jp (Y.S.); pmp61kpp@s.okayama-u.ac.jp (H.S.E.); jin@md.okayama-u.ac.jp (H.N.); 3Department of Oral and Maxillofacial Surgery, Graduate School of Medicine, Dentistry and Pharmaceutical Sciences, Okayama University, Okayama 700-8525, Japan; pikp4cqt@s.okayama-u.ac.jp; 4Department of Oral and Maxillofacial Radiology, Faculty of Medicine, Dentistry and Pharmaceutical Sciences, Okayama University, Okayama 700-8525, Japan; takeshita@okayama-u.ac.jp (Y.T.); asaumi@md.okayama-u.ac.jp (J.A.); 5Department of Oral and Maxillofacial Radiology, Okayama University Hospital, Okayama 700-8558, Japan; tomi@md.okayama-u.ac.jp; 6Department of Dental Informatics, Faculty of Medicine, Dentistry and Pharmaceutical Sciences, Okayama University, Okayama 700-8525, Japan

**Keywords:** oral squamous cell carcinoma, CXCR4, cisplatin, antitumor vascular therapy

## Abstract

**Simple Summary:**

Some cases of oral cancer are inoperable, and some of these cases also show a limited response to cisplatin. The chemokine receptor CXCR4 has been shown to be involved in tumor growth and metastasis via tumor angiogenesis, and its expression on oral squamous cell carcinoma (OSCC) cells was associated with recurrence and lymph node metastasis. Therefore, we investigated the effects of adding the CXRC4 inhibitor AMD3100 to cisplatin on OSCC cells in vitro and in mouse xenograft models in vivo. The addition of AMD to cisplatin had no additional anti-tumor effect in vitro, but improved the anti-tumor effect of cisplatin and reduced the number of CXCR4-positive blood vessels in cisplatin-resistant OSCC xenografts in vivo. These findings suggest that the addition of a CXCR4 inhibitor may increase the anti-tumor effects of cisplatin in patients with refractory OSCC.

**Abstract:**

Cisplatin is a platinum-based compound that is widely used for treating inoperable oral squamous cell carcinoma (OSCC) in Japan; however, resistance to cisplatin presents a challenge and innovative approaches are required. We aimed to investigate the therapeutic potential of targeting the chemokine receptor CXCR4, which is involved in angiogenesis and tumor progression, using the CXCR4 inhibitor AMD3100, in combination with cisplatin. AMD3100 induced necrosis and bleeding in OSCC xenografts by inhibiting angiogenesis. We investigated the combined ability of AMD3100 plus cisplatin to enhance the antitumor effect in cisplatin-resistant OSCC. An MTS assay identified HSC-2 cells as cisplatin-resistant cells in vitro. Mice treated with the cisplatin-AMD combination exhibited the most significant reduction in tumor volume, accompanied by extensive hemorrhage and necrosis. Histological examination indicated thin and short tumor vessels in the AMD and cisplatin–AMD groups. These results indicated that cisplatin and AMD3100 had synergistic antitumor effects, highlighting their potential for vascular therapy of refractory OSCC. Antitumor vascular therapy using cisplatin combined with a CXCR4 inhibitor provides a novel strategy for addressing cisplatin-resistant OSCC.

## 1. Introduction

Surgery is the first choice for treating oral cancer, but because the oral cavity is deeply involved in facial appearance, speech, and eating, resection can cause a decline in quality of life. It is desirable to shrink the tumor before surgery or to suppress its progression until the surgery, and chemotherapy and radiation therapy are used as other treatment methods. Cisplatin, one of the main chemotherapy drugs, is a platinum-based anticancer drug that suppresses DNA synthesis and inhibits cell division by inducing DNA interstrand cross-linking. In some cases, drug resistance to cisplatin can be an obstacle to anti-cancer treatment. It has been reported that the mechanism of drug resistance to cisplatin is multifactorial, including intracellular accumulation, cytoplasmic detoxification, and DNA repair mechanisms, but the details are unclear. Drugs with tumor suppression mechanisms that are different to cisplatin are needed.

CXCR4 is a well-known chemokine receptor expressed on the plasma membrane of hematopoietic cells, which plays a crucial role in the migration of these cells from the bone marrow to blood vessels [1,2,3]. In addition, the CXCR4-stromal cell-derived factor 1 axis contributes to the formation of distant organs during embryogenesis and is a recognized pathway regulating immune responses and angiogenesis [4,5,6]. This axis has also been implicated in tumor invasion and metastasis in various cancers, including breast cancer [7,8,9], and has been associated with tumor growth and survival through participation in angiogenesis in glioblastoma and pancreatic cancer, emerging as a novel tumor angiogenesis pathway [10,11,12,13]. Notably, numerous studies have reported the expression of CXCR4 in tumor cells in oral squamous cell carcinoma (OSCC), a prevalent form of oral cancer, especially in cases with CXCR4/stromal cell-derived factor 1 expression, linking it to recurrence and lymph node metastasis [14,15].

Despite extensive research on the functions of CXCR4 expressed in tumor cells, limited attention has been paid to CXCR4 expressed in the stroma in OSCC, and no reports have confirmed its expression specifically in tumor blood vessels. We therefore investigated CXCR4 expression in the stroma of OSCC from the perspective of oral pathology, focusing on the abundance of CXCR4-positive tumor blood vessels. Treatment of OSCC tumor bearing mice with the CXCR4 inhibitor AMD3100 revealed that CXCR4 inhibition induced tumor necrosis by suppressing intratumoral angiogenesis, suggesting the potential for molecular therapy targeting CXCR4-positive tumor blood vessels [16].

Previous studies indicated that, although AMD3100 inhibited intratumoral angiogenesis and caused tumor necrosis, its impact on tumor cells themselves was weak [14,16]. We therefore proposed that combining AMD3100 with agents such as cisplatin, which targets tumor cells directly, could enhance the antitumor effect. Cisplatin is commonly used anticancer drug for treating patients with advanced, inoperable oral cancer in Japan; however, cisplatin alone shows limited efficacy in some cases of OSCC, indicating the need to explore combination therapies targeting alternative pathways [17,18,19,20]. The current study aimed to investigate the therapeutic potential of cisplatin in combination with the CXCR4 inhibitor AMD3100 for treating cisplatin-resistant OSCC. The results warrant further studies to determine if AMD3100-induced intratumoral angiogenesis inhibition affects cisplatin penetration.

## 2. Materials and Methods

### 2.1. Cell Lines and Cell Culture

We obtained human OSCC cell lines, HSC-2 and SAS, from the Japan Research Bioresource Cell Bank (JCRB) at the National Institutes of Biomedical Innovation, Health and Nutrition (Osaka, Japan). The cells were cultured in Dulbecco’s modified Eagle’s medium (Sigma-Aldrich, St. Louis, MO, USA) supplemented with 10% fetal bovine serum (Thermo Fisher Scientific, Waltham, MA, USA) and 100 U/mL penicillin and 100 µg/mL streptomycin (Sigma-Aldrich) at 37 °C in humidified air with 5% CO_2_.

### 2.2. Reagent

AMD3100/plerixafor (ChemScene, Monmouth Junction, NJ, USA) (10 mg) was dissolved in 20 mL of physiological saline and stored at 4 °C before use. The same concentration of drug was used in the experiments in mice.

### 2.3. Cell Proliferation Assay (MTS Assay)

The MTS assay was performed using The CellTiter 96^®^ AQueous One Solution Cell Proliferation Assay (Promega, Madison, WI, USA). HSC-2 and SAS cells were seeded at 5 × 10^3^ cells per well in 96-well plates. After culture for 24 h, 2.5 µg/mL cisplatin, 1 µg/mL AMD3100, or a mixture of 2.5 µg/mL cisplatin plus 1 µg/mL AMD3100 was added. After further incubation for 0, 24, or 48 h, the MTS reagent was added for 1 h and the absorbance at 490 nm was then measured.

### 2.4. Xenografts and Administration of AMD3100 and Cisplatin to Mice

HSC-2 or SAS cells, 1 × 10^6^ cells per xenograft, were transplanted subcutaneously into the backs of BALB/c-nu/nu nude mice. After 14 days, the mice were divided into four groups: a control group, AMD group, cisplatin group, and AMD + cisplatin group (*n* = 5 per group). AMD3100 (50 µg/day) was administered intraperitoneally to mice in the AMD and AMD + cisplatin groups for 21 days, and cisplatin was administered intraperitoneally to mice in the cisplatin and AMD + cisplatin groups every week for 21 days. Control mice received xenografts as above, followed by intraperitoneal saline. The largest tumor diameter (L) and smallest tumor diameter (W) were measured on the day the drug administration started, and one week, two weeks, and three weeks later, and the tumor volume (V) was calculated (V = L × W^2^). Mice were sacrificed and perfused with 10% neutral buffered formalin. Tumors along with surrounding tissues were excised, immersion fixed in 10% neutral buffered formalin, dehydrated, embedded in paraffin, and sectioned to prepare slides. The slides were then stained with hematoxylin and eosin (HE) or immunohistochemistry (IHC) and examined under a light microscope.

### 2.5. Immunohistochemistry (IHC)

We performed IHC using anti-CD34 antibody (Histofine, Tokyo, Japan, undiluted). The signal was enhanced by the avidin–biotin complex method (Vectastain ABC Kit, Vector Laboratories, Newark, CA, USA). Detection was performed with 3,3′-diaminobenzidine, and the specimens were observed under a light microscope. Comparison of tumor vasculature was performed using the largest tumor sections and analyzed with ImageJ software 1.54a.

### 2.6. Statistical Analysis

Data were recorded using Microsoft Excel (Microsoft, Inc., Redmond, WA, USA) and entered into an electronic database. Normally distributed data were presented as means and standard deviations. The database was transferred to IBM SPSS Statistics version 27 (Chicago, IL, USA) for statistical analysis to investigate the differences between the group means. Data were analyzed by *t*-test, with a significance level of *p* < 0.05.

## 3. Results

### 3.1. MTS Assay

Cisplatin alone inhibited the survival and proliferation of HSC-2 and SAS cells (Figure 1), while AMD3100 alone had no effect on the survival or proliferation of either cell type. AMD alone appeared to slow the proliferation of SAS cells at 24 and 48 h compared to saline, but the increase in absorbance was not statistically significant. The lack of effect of AMD3100 was consistent with previous reports [14]. AMD + cisplatin suppressed tumor cell survival and proliferation to the same extent as cisplatin alone, indicating that AMD3100 did not inhibit the tumor cell-suppressive effect of cisplatin.

### 3.2. Administration of AMD3100 and Cisplatin to Tumor Cell-Implanted Mice

#### 3.2.1. Histological Features of HSC-2 and SAS Cells

Both HSC-2 and SAS are human-derived squamous cell carcinoma cell lines, but their histological characteristics differ (Figure 2). HSC-2 showed well-differentiated features, while the SAS cells showed poorly differentiated features. These findings indicated that the characteristics of the respective cell lines were maintained.

#### 3.2.2. Tumor Volume

Among HSC-2-xenografted mice, the group treated with AMD + cisplatin had the smallest tumors 3 weeks after drug administration (Figure 3a). There was no obvious tumor suppression in the cisplatin group, but the increase in tumor volume was significantly smaller in the AMD + cisplatin group from 2 to 3 weeks after drug administration (Figure 3b). In SAS-xenografted mice, no differences in tumor volume were observed between the cisplatin, AMD, and AMD + cisplatin groups (Figure 3c,d).

#### 3.2.3. Histological Findings of Resected Tumors

Histological images of tumor tissues excised 3 weeks after the start of drug administration are shown in Figure 4. The drug administration schedule is shown in Figure 4a. We focused on the pattern of necrosis in the tumor. Tumors in HSC-2-transplanted mice treated with saline or cisplatin showed extensive necrosis in the center of the tumor and viable tumor cells around the necrosis (Figure 4b,c), while mice in the AMD and AMD + cisplatin groups showed necrosis in the center of the tumor and minute hemorrhage and necrosis at the margin of the tumor occupied by viable tumor cells (Figure 4d,e). AMD3100 thus affected the pattern of intratumoral necrosis in HSC-2-implanted mice. In contrast, large and small areas of necrosis were observed within the tumor in all four SAS-implanted groups, at both the center and margin. AMD3100 thus had no effect on the necrosis or tumor histology in SAS-transplanted mice (Figure 4f,g).

#### 3.2.4. Comparison of Intratumoral Blood Vessels

The intratumoral blood vessels in the HSC-2-engrafted mice were long and branched (Figure 5a), while those in the SAS-transplanted mice were short and poorly branched (Figure 5b). There was no significant difference in the number of CXCR4-positive vessels between the saline-treated HSC-2 and SAS tumors (Figure 5c), while AMD3100 reduced the number of CXCR4-positive vessels in the HSC-2 tumors (Figure 5d), but not in the SAS tumors (Figure 5e).

We examined the blood vessels using the vascular marker CD34. HSC-2 blood vessels were distributed abundantly, including some >100 µm, and were accompanied by branching. There were no differences in the distribution or morphology of blood vessels in HSC-2 tumors between the saline and cisplatin groups (Figure 6a,b), while the blood vessel distribution was decreased and the vessels were shortened in the AMD and AMD + cisplatin groups, and the blood vessels were also poorly branched (Figure 6c,d). The blood vessels in the SAS tumors were distributed abundantly, with most specimens having short blood vessels (≤50 µm), and the blood vessels were scarcely branched. There were no differences in the distribution or morphology of the blood vessels in SAS tumors between the saline and cisplatin groups (Figure 6e,f), or between the AMD and AMD + cisplatin groups (Figure 6g,h).

#### 3.2.5. Comparison of Intratumoral Blood Vessel Length

In HSC-2-transplanted mice, the intratumoral blood vessels were shorter in the AMD + cisplatin group compared with the cisplatin group (Figure 7). Among the four groups, tumors in the AMD + cisplatin group had the highest percentage of blood vessels of ≤50 µm. There was no difference in the lengths of intratumoral blood vessels among the four SAS-implanted groups.

## 4. Discussion

The current study found no discernible impact of AMD3100 on tumor cells in vitro, and no significant difference in efficacy between cisplatin with and without AMD3100. In contrast, however, AMD3100 demonstrated an antitumor effect on HSC-2 xenografts in vivo. Cisplatin or AMD3100 alone showed distinct patterns of tumor necrosis, while the combination of the two drugs resulted in both patterns, suggesting a potential synergistic interaction enhancing the antitumor effect. Notably, the response to AMD3100 varied between OSCC cell lines, with HSC-2 (moderately to highly differentiated) tumors displaying different vascular characteristics to SAS (poorly differentiated) tumors. HSC-2-implanted mice exhibited thick, long, and branched blood vessels, and AMD3100-induced inhibition of CXCR4 expression, while the SAS-implanted mice displayed thin and short vessels, with unclear AMD3100-induced changes in CXCR4 expression. AMD + cisplatin resulted in the most effective tumor suppression in HSC-transplanted mice, whereas cisplatin and AMD + cisplatin had no effect on tumor volume in SAS-transplanted mice. This discrepancy might be attributed to the weaker impact of AMD3100 on poorly differentiated OSCCs, such as SAS, resulting in insufficient efficacy in combination with cisplatin. These results emphasize the fact that, although CXCR4 suppression has no direct effect on tumor cells, it induces an antitumor response in vivo, particularly in cases with abundant CXCR4-positive blood vessels. This underscores the influence of CXCR4 on the stroma, notably by inhibiting intratumoral blood vessel formation. Because this study used human tumor cell lines in vivo, immunodeficient mice were used to prevent rejection. Therefore, immune cell infiltration should not be considered. However, the role of immune cells in the tumor microenvironment is a very interesting field, and to fully elucidate the effects of CXCR4 inhibitors on tumors, the effects of CXCR4 inhibitors on the tumor immune system should be examined. This is a topic we would like to explore in the future.

Surgical excision remains the primary treatment for oral cancer, but associated deteriorations in quality of life may necessitate alternative approaches. Although cisplatin is commonly used as a non-surgical anticancer treatment, its efficacy as a standalone treatment varies. The combination of cisplatin with a CXCR4 inhibitor offers potential benefits including tumor size reduction, minimizing resection-site impact, and enhancing patient quality of life. Clinical practice currently uses molecular therapy targeting vascular endothelial growth factor (VEGF) to inhibit the tumor vasculature in cancers including colorectal, lung, and breast cancer [21,22,23]; however, its efficacy is limited by a short treatment response period and the development of drug resistance [24,25,26,27,28]. Activation of CXCR4 increases VEGF expression and promotes angiogenesis. This action contributes to the drug resistance of bevacizumab as it provides more VEGF-to-VEGF inhibitors [29]. Furthermore, CXCR4 is known to be involved in cell proliferation and angiogenesis in the MEK-pERK pathway, apart from the PGK-AKT pathway, which is activated by VEGFR, and also induces angiogenesis in a pathway separate from VEGF [30]. The CXCR4 angiogenesis pathway differs from VEGF, making cisplatin–CXCR4 inhibitor combination therapy a novel approach for anti-tumor-vascular treatment. The CXCR4 inhibitor AMD3100 has been approved by the US Food and Drug Administration and European Medicines Agency and has demonstrated no serious side effects in clinical applications mobilizing hematopoietic stem cells [31,32]. The potential antitumor effects of AMD3100 demonstrated in this study highlight its clinical potential in combination with existing anticancer and molecular targeted drugs, and as a new treatment option for patients resistant to VEGF inhibitors. In a previous report, when the antitumor effect of VEGF on HSC-2 and SAS was examined in vivo and in vitro, VEGF inhibitors alone could not inhibit tumors, but when combined with other anticancer drugs, a synergistic antitumor effect was observed [33]. This suggests that VEGF inhibition does not directly attack tumors, but indirectly exerts an antitumor effect by approaching tumor blood vessels, which is similar to the results of the current study using CXCR4 inhibitors. The effect of combining CXCR4 inhibitors and VEGF inhibitors on tumors is interesting and should be investigated in the future.

## 5. Conclusions

The findings of this in-vivo and in-vitro study suggest the potential efficacy of combining AMD3100 with cisplatin for treating OSCC in cases where the tumor-suppressive effect of cisplatin alone is limited. The amalgamation of cisplatin and a CXCR4 inhibitor may provide a promising novel approach for managing refractory OSCC.

## Figures and Tables

**Figure 1 cancers-16-02326-f001:**
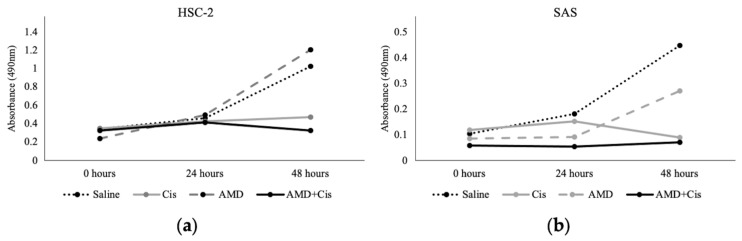
MTS assay. Measurement of the effect of drugs on cancer cells proliferation in oral squamous cell carcinoma cell lines (**a**) HSC-2 and (**b**) SAS cells. The cisplatin, AMD3100, and mixture of cisplatin and AMD3100 drugs were added separately to each OSCC cell line at incubation time 0 h. Tumor cell survival was compared by measuring the absorbance after 24 and 48 h.

**Figure 2 cancers-16-02326-f002:**
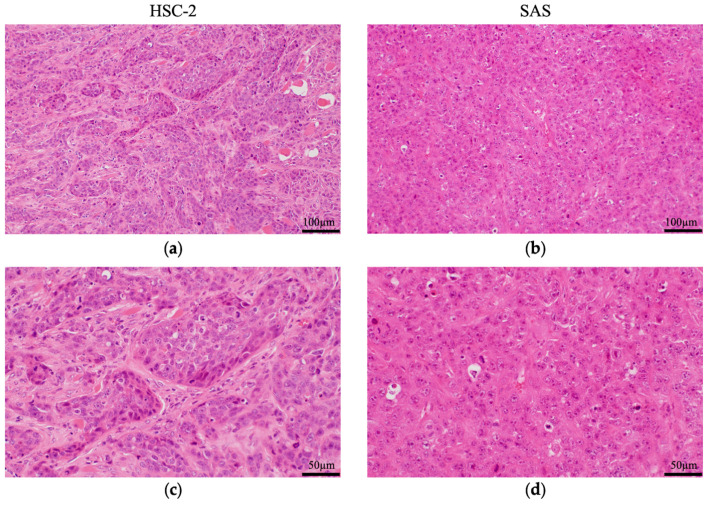
Pathological analysis of HSC-2 and SAS tumors in mice. Hematoxylin–eosin (HE) staining of resected (**a**) HSC-2 and (**b**) SAS tumors. High-magnification images of (**c**) HSC-2 and (**d**) SAS tumors. HSC-2 xenografts formed a tumor mass and tumor cells differentiated into basal cell-like and spinous cell-like cells, with fibrous connective tissue and blood vessels in the stroma. SAS xenografts did not form tumor alveoli and proliferated diffusely with necrosis. The tumor cells were poorly differentiated with poor stroma. *n* = 5.

**Figure 3 cancers-16-02326-f003:**
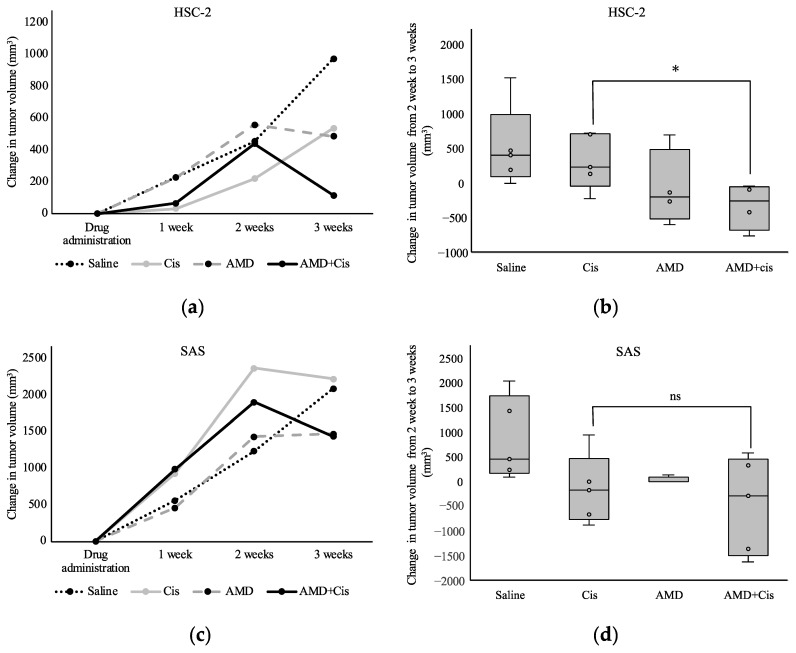
Effects of AMD3100 and cisplatin on tumor volume in mice transplanted with HSC-2 or SAS cells. Changes in volumes of (**a**) HSC-2 and (**c**) SAS tumors 3 weeks after administration of AMD3100 alone, cisplatin alone, or AMD3100 + cisplatin. Changes in volumes of (**b**) HSC-2 and (**d**) SAS tumors are measured from week 2 to week 3 after addition of cisplatin or AMD3100 + cisplatin. *n* = 5. * *p* < 0.05. ns, not significant.

**Figure 4 cancers-16-02326-f004:**
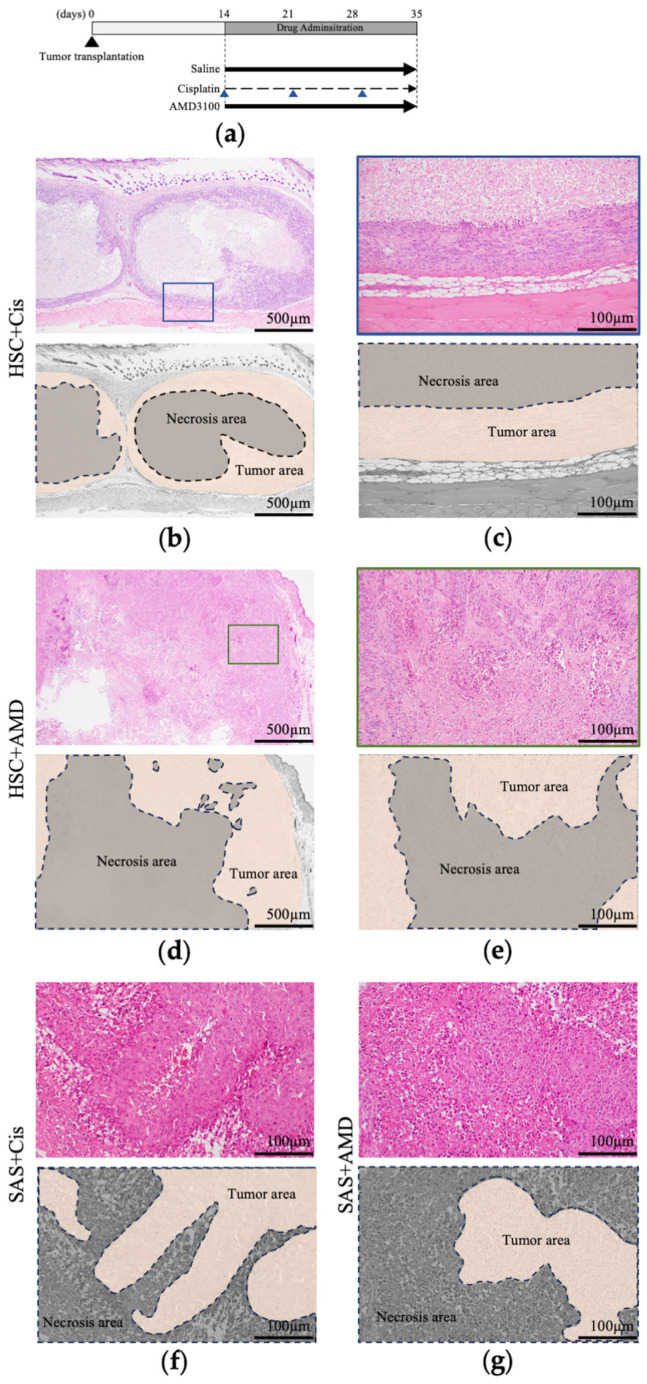
Pathological analysis of the effects of AMD3100 and cisplatin on tumor xenografts. (**a**) Schematic protocol of tumor transplantation and drug administration. Arrows represent the time points of administration. (**b**) Image of HE staining of resected HSC-2 tumor in mice treated with cisplatin. (**c**) High-magnification image of region with viable tumor cells in cisplatin-treated HSC-2 tumor. (**d**) Image of HE staining of resected HSC-2 tumor in mice treated with AMD3100. (**e**) High-magnification image of region with viable tumor cells in AMD3100-treated HSC-2 tumor. Image of HE staining of resected SAS tumors in (**f**) cisplatin and (**g**) AMD groups. *n* = 5.

**Figure 5 cancers-16-02326-f005:**
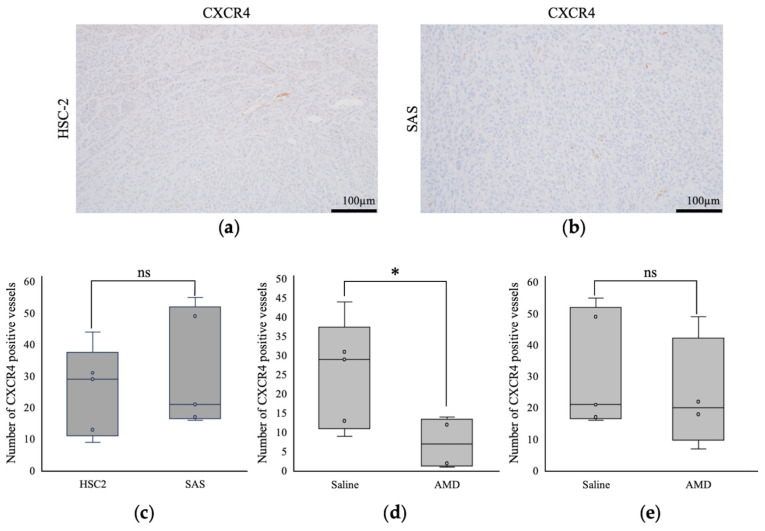
Comparison of intratumoral blood vessels by immunohistochemistry (IHC). Images of CXCR4 positive vessel structure in saline-treated (**a**) HSC-2 and (**b**) SAS tumors. Numbers of CXCR4-positive blood vessels were counted in five random locations per case in (**c**) HSC-2 and SAS saline-treated tumors, (**d**) HSC-2 saline and AMD3100-treated tumors, and (**e**) SAS saline- and AMD3100-treated tumors. *n* = 5. * *p* < 0.05. ns, not significant.

**Figure 6 cancers-16-02326-f006:**
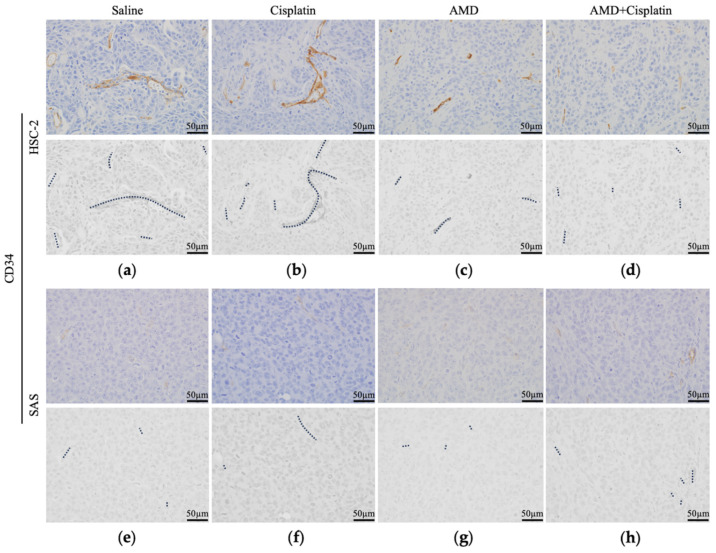
Comparison of CD34 positive intratumoral blood vessel structures by IHC. HSC-2 (**a**) saline, (**b**) cisplatin, (**c**) AMD, and (**d**) AMD + cisplatin groups. SAS (**e**) saline, (**f**) cisplatin, (**g**) AMD, and (**h**) AMD + cisplatin groups. Dotted lines represent CD34 positive vessel structures. *n* = 5.

**Figure 7 cancers-16-02326-f007:**
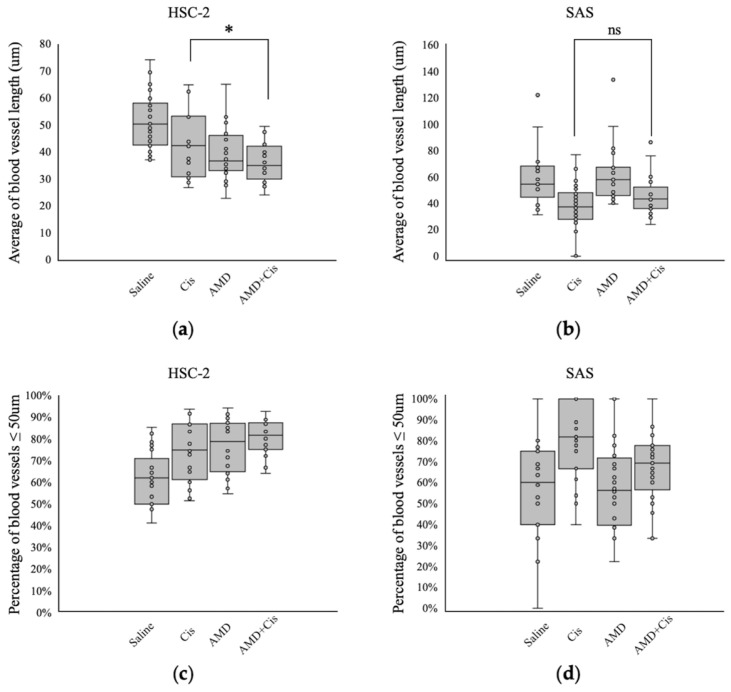
Effects of AMD3100 and cisplatin on intratumoral blood vessel length in HSC-2- and SAS-transplanted mice. Average blood vessel length in mice transplanted with (**a**) HSC-2 and (**b**) SAS cells. Percentage of blood vessels ≤ 50 µm (number of blood vessels ≤ 50 µm/total number of blood vessels multiplied by 100) in mice transplanted with (**c**) HSC-2 and (**d**) SAS cells. *n* = 5. * *p* < 0.05. ns, not significant.

## Data Availability

All other data supporting the findings of this study are available from the corresponding authors upon request.

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
