# Peer review of "Efficacy of Cisplatin–CXCR4 Antagonist Combination Therapy in Oral Cancer"

_cancers, 2024, doi:10.3390/cancers16132326_

Round 1

Reviewer 1 Report

Comments and Suggestions for Authors

Yoshida, et al. evaluated the combined ability of AMD3100 plus cisplatin to enhance the antitumor effect in cisplatin-resistant OSCC. They used two OSCC tumor models with contrasting differences in tumor vasculature to carry their studies. Authors concluded beneficial efficacy of combining AMD3100 with cisplatin for treating OSCC in cases where the tumor-suppressive effect of cisplatin alone is limited. Authors investigated important therapeutic area in OSCC that needs to be addressed as resistance is a problem in long-term therapy in cancers like OSCC.  Manuscript can benefit by addressing the following comments/drawbacks. 

1. Authors should start Introduction section about oral cancer, its therapeutic options, and drug resistance and associated mechanisms. 

2. Starting at line 61 "Treatment of OSCC model mice with..." can be changed to "Treatment of OSCC tumor-bearing mice with...".

3. At line 68, "Cisplatin is commonly used as an anticancer drug for treating patients...." can be changed to "Cisplatin is commonly used anticancer drug for treating patients....".

4. At line 71, "The current study aimed to investigate the therapeutic potential of cisplatin combined with the CXCR4 inhibitor AMD3100 for treating OSCC with an inadequate response to cisplatin alone." should be changed to "The current study aimed to investigate the therapeutic potential of cisplatin in combination with the CXCR4 inhibitor AMD3100 for treating cisplatin-resistant OSCC."

5. At line 96 (regarding HSC-2 or SAS cell numbers for xenografts), 10 × 105 cells can simply be denoted by 1 x 106 cells.

6. All the figures should be provided as high quality for more clarity. Axes and scales in graphs should be sharp and easily visible. Bold fonts will help.

7. Figure 1 lacks significance values. y-axis should be properly labelled.

8. Figure 1b data shows that AMD treatment slows growth of SAS cells compared to saline-treated cells at 24 and 48 hours. There also seems less metabolic activity in AMD-treated cells at 24 hours when compared to Cis-treated SAS cells. Authors need to check stats and discuss this results.

9. Text at line 149 "Among SAS-xenografted mice, cisplatin alone suppressed the tumor volume (Figure 3c),..." does not match Figure 3c (indeed Cis group shows more tumor volume than Saline group). Authors need to clarify this. Also, Figure 2 panels should denote cell lines at top. Layman readers don't like to go back to text/legends to check which panel is for what cell line.

10. IHC panels should indicate x magnification/size in microns at the bottom of each photo.

11. Check the y-axis for Figure 3a. cm3?

12. Figure legends should be having brief details about the method.

13. Provide the number of samples (n)/group in the figure legends.

14.  Figure 3 legends is confusing. a and c show changes at wk1, wk2 and wk3. However, b and d shows changes at one time-point. Make it clear in the legend.

15. Figure 4 panels are miss-labelled for numbers. Also these should be labelled for tumor type and treatment.

16. You can show the magnification, without labelling the panel, by drawing a downward arrow from the region of the panel it originates from. 

16. What is the status of immune-cell infiltration into these tumors with various treatments. Can you show the data for T-cell infiltration into tumors.  Also, it seems that AMD3100 blocks tumor vasculature for its efficacy, do you have data for VEGF blocking in these tumors alone or in combination. Or you can discuss this area if you don't have the data.

17. Authors should correct typos in the text. Check the references style for uniformity.

Comments on the Quality of English Language

Minor editing needed.

Reviewer 2 Report

Comments and Suggestions for Authors

this is an interesting work, and the paper is well written. however, the authors are encouraged to consider the following points. 

- please report ethical clearance for the use of animals in this study. 

- it is not clear if the mice in all groups had similar/comparable tumor sizes before the start of the treatment. kindly show data if available. if the mice have been randomized according to tumor size please state that clearly.

- Figure 5 and 6 have a bit low quality and seems out of focus, please replace with better quality figures. 

- Discussion is a bit short and does not present some important aspects. for example: how would this combined treatment differ from convensional anti-angiogenic drugs, which has been disappointing at clinical level. it is also interesting that the effect was marked in one of the two cell lines used but not in both, how would you investigate selection criteria of tumors that might benifit of this combined treatment. also, CXCR4 is expressed on inflammatory cells, which might have an effect on tumor development and growth, but the mouse strain used in this study is immunodeficient, is possible that this had an effect on the results of the study ? would the outcome be different in another strain ? or a syngeneic model (mouse cancer cell line in immunocompitent animals) ?. are the results applicable for other tumor types ? kindly consider discussing these points. 
